

# Variability of soil carbon and nitrogen stocks after conversion of natural forest to plantations in Eastern China

Mbezele Junior Yannick Ngaba, Xiang-Qing Ma and Ya-Lin Hu

Forest Ecology and Stable Isotope Research Center, College of Forestry, Fujian Agriculture and Forestry University, Fuzhou, Fujian, People's Republic of China

## ABSTRACT

Forest plantation, either through afforestation or reforestation, has been suggested to reverse and mitigate the process of deforestation. However, uncertainties remain in the potential of plantation forest (PF) to sequestrate carbon (C) and nitrogen (N) compared to natural forest (NF). Soil C and N stocks require a critical and updated look at what is happening especially in the context of increasing rate of land use change and climate change. The current study was conducted in China's Eastern forest to estimate soil C and N stocks in six depth layers (0–10, 10–20, 20–40, 40–60, 60–80 and 80–100 cm) and two forest types (NF and PF) at four sites along climate factors gradient. The results showed that the overall mean soil C and N amounts to a depth of 20 cm ranged from $2.6 \pm 1.1$ Mg ha$^{-1}$ to $38.6 \pm 23.1$ Mg ha$^{-1}$, and soil nitrogen stock ranged from $0.2 \pm 0.1$ Mg ha$^{-1}$ to $3.3 \pm 1.5$ Mg ha$^{-1}$. Moreover, a loss of C stock was observed at Qingyuan (QY) by –7%, Dinghushan (DH) by –26%, Jianfengling (JF) by –13% while that of N stock was observed at QY (–8%), DH (–19%) and JF (–12%) at both depth layers. These results indicate that NFs have a better capacity to accumulate soil C and N. The soil C and N decreased from the southeast to the northeast and increased from tropical to temperate mixed forests zone in the eastern part of the study area. The C and N stock mainly occurred in the topsoil and decreased significantly with depth. Moreover, soil C and N stocks increased with age of plantation. This study provides an overview of the current spatial distribution and soil stocks of C and N, as well as the effects of environmental factors on soil C and N stocks. It also indicated that, although mean annual temperature and mean annual precipitation are the key factors affecting the variations in soil C and N, their vertical and horizontal distribution differed in various aspects.

## INTRODUCTION

Soil is the major carbon sink of global terrestrial ecosystems, storing about 1,550 Pg of carbon (C), which is twice the atmospheric carbon pool (*Lal, 2004*). The C stock is closely linked to the soil nitrogen (N) which could increase soil C stocks through promoting plant growth and improving the net primary productivity of terrestrial ecosystem (*Solberg et al., 2004*). However, it is limited by the accessibility of soil N in forest ecosystem due to the fact that N dynamics can regulate terrestrial carbon sequestration, for example,

Corresponding author
Ya-Lin Hu, huyl@iae.ac.cn

increasing N inputs leads to sustainable C sequestration (*Reich & Oleksyn, 2004*; *Deng et al., 2016*). Soil N does not only have a substantial impact on soil carbon sinks through the interaction between soil nitrogen and carbon, but also on maintaining the ecological function of plantation ecosystem (*Reich et al., 2006*; *Liu et al., 2016*). Furthermore, the C stability of terrestrial ecosystems is notably sensitive and can be directly affected to impact human activities in the short term, including deforestation, biomass burning, land-use changes, forest management practices and environmental pollution (*Batjes & Dijkshoorn, 1999*; *Stockmann et al., 2013*). It has been recognized that small fluctuations of soil organic carbon (SOC) pool could have large impacts on the atmospheric carbon dioxide ($CO_2$) concentration by implication in the control of the greenhouse effect (*Lal, 2004*; *Powlson, Whitmore & Goulding, 2011*).

Due to the importance of soil C and N as a source or as a sink to atmospheric $CO_2$, several studies have assessed soil C and N stocks (*Batjes, 2002*; *Maquere et al., 2008*; *Gross et al., 2018*; *Gao et al., 2019b*; *Li et al., 2019*; *Da Silva Santana et al., 2019*). Besides, an assessment of soil C stock is crucial for the evaluation of the capacity of soil to sequester atmospheric C. It has been demonstrated that soil C and N stocks are influenced by the complex interactions of tree species, management, climate, vegetation cover, land use (LU), bulk density (BD), soil type and texture (*Batjes, 1996*; *Lal, 2005*). For example, numerous studies have reported that tree species can alter soil C and N stocks by many processes such as changes in litter quantity and quality, turnover rate of roots and exudates, microbial communities and soil physicochemical properties (*Paul et al., 2002*; *Pérez-Cruzado et al., 2012*; *Wang et al., 2013*; *Hoogmoed et al., 2014*; *Deng & Shangguan, 2017*). Globally, LU change is the second largest source of greenhouse gas emissions (GHG) after burning fossil fuels and the first for the tropical region (*Houghton, 2003*). Deforestation through the conversion of natural forests (NF) to other LUs contributes to GHGs emission and could alter soil C and N cycles by variations in the amount of forest floors and may have a significant impact on the total amount of GHG emissions (*Minasny et al., 2017*; *Gao et al., 2019a*). Ensuring the stability of forest carbon stocks is an essential global challenge in the future. Instituting control measures to reduce GHG from forests is therefore essential for maintenance of stable stock of forest carbon.

Plantation forest (PF) either through afforestation or reforestation have been suggested to overcome this problem and ensure the sustainability of the forests through the process of deforestation, which consequently favors the accumulation of C (*Metz et al., 2007*) as PF have potential to contribute to "Kyoto Protocol" targets for reducing net national GHG emissions (*Bolin, 1998*) as large quantities of atmospheric $CO_2$ can be fixed into tree biomass for a long period of time (*Hoen & Solberg, 1994*). Although, the assessment of the soil C and N stock is complicated, it is crucial to assess if all PF necessarily result in carbon sequestration in the soil or whether this sequestration is significant. In other words, it is essential to know if it can play the same role as PF of storing soil carbon. Therefore, quantitative assessment of soil C and N stocks and their dynamics is crucial in understanding the carbon sink capacity of terrestrial ecosystems in the context of the climate change. We hypothesized that the conversion of NF–PF would result in a loss of C and N stocks, and that topsoil has higher potential accumulation than subsoil in both

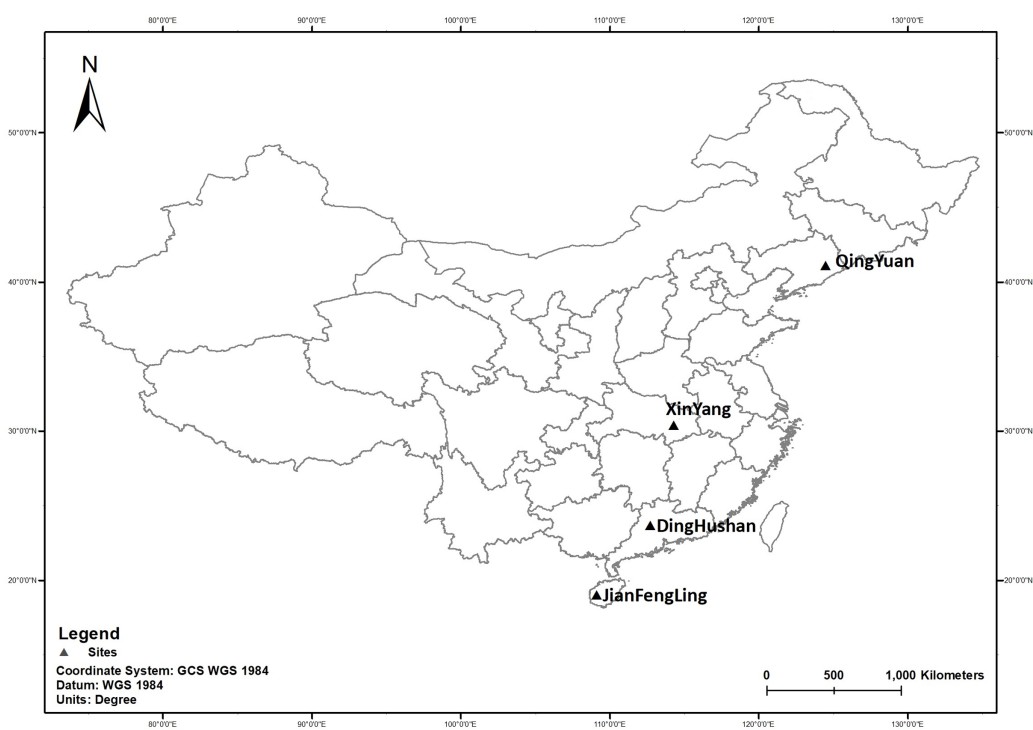

**Figure 1 The location of forest stands at four sites across eastern China.** QY, Qingyuan; HT, Huitong; DH, Dinghushan; JF, Jianfengling.

forest types. Also, we expected that C, N stocks will decline with depth, but would be influenced by mean annual temperature (MAT) and mean annual precipitation (MAP) especially in NF. Therefore, the objectives of this study were to (1) estimate the soil C and N stocks within the 0–100 cm soil layer along and compare their variations throughout the profile under different LU types; (2) investigate the effects of forest conversion in soil C and N stocks and assess the interaction between soil C and N changes and (3) analyze the effects of environmental factors and main factors influencing soil C and N stocks.

## MATERIALS AND METHODS

### Study site and soil sampling

The study sites were located in Chinese eastern forest which extends from Hainan Island to China's northern border, ranging from 105°E to 130°E at latitudes and from 10°N to 50°N (Fig. 1). This zone provides an ideal platform to research carbon and nitrogen of forest ecosystems in East Asia's monsoon region (*Yu et al., 2008*). Along the Eastern forest of China, we chose four forests which included four forest sites were selected as experimental plots including Qingyuan (QY), Huitong (HT), Dinghushan (DH) and Jianfengling (JF) across eastern China forest (Fig. 1). Vegetation sequence distribution includes temperate mixed forests, evergreen broadleaved forest, subtropical evergreen coniferous forest and tropical rainforest from north to south (*Sheng et al., 2014*).

The specific characteristics of the sampling sites are described in Table 1. A total of 168 samples were collected in August 2017 from NF to PF stand of each site. Three pits were

**Table 1 Location and characteristics of forest stands at four sites across the eastern China.**

| Sites | Location | pH | | Province | Elevation (m) | Soil type | Climate zones | MAT (°C) | MAP (mm) | Natural land use | Plantation type | Plantation age |
|---|---|---|---|---|---|---|---|---|---|---|---|---|
| | | NF | PF | | | | | | | | | |
| Qingyuan (QY) | N 41.85 E 124.93 | 5.57 | 5.54 | Liaoning | 597 | Brown forest soil | Mid temperate | 5.91 | 794 | DBL | *Pinus koraiensis.* | 38 |
| Huitong (HT) | N 26.85 E 109.60 | 4.58 | 4.69 | Hunan | 427 | Lateritic red soil, yellow soil | Mid subtropical | 17.17 | 1,256 | EBL | *Cunninghamia lanceolata* | 33 |
| Dinghushan (DH) | N 23.17 E 113.52 | 4.02 | 4.37 | Guangdong | 275 | Lateritic red soil, yellow soil | Southern subtropical | 21.08 | 1,955 | MEB | *Pinus massoniana* | 30 |
| Jianfengling (JF) | N 18.44 E 108.01 | 4.52 | 4.69 | Hainan | 800 | Yellow soil | Tropical | 19.80 | 2,499 | TMF | *Pinus caribaea* | 30 |

Note:
DBL, Deciduous broad-leaved forest; EBL, Evergreen broad-leaved forest; MEB, Monsoon evergreen broad-leaved forest; TMF, Tropical monsoon forest. NF, Natural Forest; PF, Plantation Forest.

dug at one m depth for soil sampling and mineral soil samples were collected at six depth layers 0–10, 10–20, 20–40, 40–60, 60–80 and 80–100 cm along soil profiles. MAT and MAP data collected from 1960 to 2014 in the adjacent climate monitoring stations are utilized herein. Soil C and N content were measured using an isotope ratio mass spectrometer (IsoPrime 100; Isoprime Ltd., Cheadle, UK), connected to a CN elemental analyzer (Vario MICRO cube; Elementar, Langenselbold, Germany). Soil pH was determined using a digital potentiometric pH meter in 1:3 soil suspensions in both 0.01 M $CaCl_2$ solution and deionized water. Soil samples were extracted using a stainless-steel cylinder of 100 $cm^3$ in volume of the undisturbed soil samples and BD was calculated by dividing the oven-dried weight of fine earth by the volume of the core (Table 2).

## Statistical analysis

Soil C and N stocks were determined on a weight to area basis (kg of C/N per $m^2$ of soil) per soil depth class was calculated using the following formula:

$$TX = \sum_{i}^{n} X\%_i \times BD_i \times v_i \times a$$

where TX is soil C or N stock (kg C $m^{-2}$); $X\%_i$ is C or N in percentage at depth $i$; $BD_i$ is bulk density at depth $i$; and $v_i$ is volume of soil at each horizon; $a$ is the instrument's typical precision ±0.005% for C and ±0.001% for N according to the manufacturer's standard material (Assefa et al., 2017). The carbon and nitrogen content data were obtained from our previous manuscript (Ngaba et al., 2019) (Table 3).

Two-way ANOVA method was used to test the significance of differences in site, depth and their interactions on soil C and N stocks under NF and PF using a significance level of $\alpha = 0.05$. Pearson correlation analysis was used to analyze the relationships between

**Table 2 Soil C and N content, Soil BD under different forest types.**

| | Soil depth (cm) | QY | | HT | | JF | | DH | |
|---|---|---|---|---|---|---|---|---|---|
| | | NF | PF | NF | PF | NF | PF | NF | PF |
| C content (g kg$^{-1}$) | 0–10 | 48.95 | 41.06 | 17.09 | 15.15 | 25.38 | 19.39 | 25.77 | 14.67 |
| | 10–20 | 26.67 | 23.41 | 10.03 | 12.87 | 15.83 | 12.48 | 12.53 | 5.05 |
| | 20–40 | 12.84 | 12.99 | 8.54 | 9.43 | 11 | 6.93 | 6.97 | 3.93 |
| | 40–60 | 6.97 | 5.82 | 5.47 | 6.02 | 6.58 | 6.21 | 5.14 | 3.03 |
| | 60–80 | 4.38 | 3.86 | 4.54 | 5.07 | 4.61 | 8.52 | 4.7 | 6.21 |
| | 80–100 | 2.76 | 3.07 | 4.06 | 4.31 | 3.34 | 3.28 | 3.57 | 3.41 |
| N content (g kg$^{-1}$) | 0–10 | 4.23 | 3.23 | 1.99 | 1.66 | 1.98 | 1.51 | 2.13 | 1.24 |
| | 10–20 | 2.6 | 2.42 | 1.3 | 1.48 | 1.32 | 1.13 | 1.13 | 0.6 |
| | 20–40 | 1.33 | 1.49 | 1.18 | 1.17 | 0.97 | 0.64 | 0.75 | 0.49 |
| | 40–60 | 0.79 | 0.69 | 1 | 0.97 | 0.66 | 0.64 | 0.67 | 0.45 |
| | 60–80 | 0.51 | 0.49 | 0.94 | 0.91 | 0.5 | 1.17 | 0.63 | 1.39 |
| | 80–100 | 0.34 | 0.36 | 0.92 | 0.9 | 0.36 | 0.35 | 0.57 | 0.56 |
| BD (g cm$^{-3}$) | 0–10 | 0.80 | 0.95 | 1.10 | 1.26 | 0.98 | 1.08 | 0.98 | 1.54 |
| | 10–20 | 1.03 | 1.09 | 1.12 | 1.21 | 1.14 | 1.31 | 1.30 | 1.66 |
| | 20–40 | 1.27 | 1.26 | 1.13 | 1.28 | 1.24 | 1.48 | 1.41 | 1.69 |
| | 40–60 | 1.27 | 1.32 | 1.16 | 1.26 | 1.34 | 1.41 | 1.39 | 1.51 |
| | 60–80 | 1.28 | 0.92 | 1.18 | 1.37 | 1.54 | 1.36 | 1.52 | 1.48 |
| | 80–100 | 1.20 | 0.76 | 1.25 | 1.33 | 1.43 | 1.41 | 1.46 | 1.55 |

Notes:
Data are means of three plots.
NF, Natural forest; PF, Plantation forest; QY, Qingyuan; HT, Huitong; DH, Dinghushan; JF, Jianfengling; BD, Bulk density.

**Table 3 Two-way ANOVA results for all soil variables in both forest type.**

| | | NF | | PF | |
|---|---|---|---|---|---|
| | Variables | F | P | F | P |
| Site | C stock | 15.16 | *** | 3.31 | ** |
| | N stock | 28.99 | *** | 4.85 | ** |
| Depth | C stock | 5.35 | ** | 36.33 | *** |
| | N stock | 4.17 | n.s | 36.39 | *** |
| Site * Depth | C stock | 0.31 | n.s | 2.24 | n.s |
| | N stock | 0.32 | n.s | 2.71 | n.s |

Notes:
$n = 15$ (Depths), $n = 4$ (Site).
* Indicate a significant level at $P < 0.05$.
** Indicate a significant level at $P < 0.01$.
*** Indicate a significant level at $P < 0.001$, respectively.
n.s, non significant.

soil C or N stocks and pH, BD, elevation, age of plantation, MAT and MAP. All statistical analyses were performed using the SPSS version 20.0 (Systat Statistical Software Package for Windows) (*Coakes & Steed, 2009*).

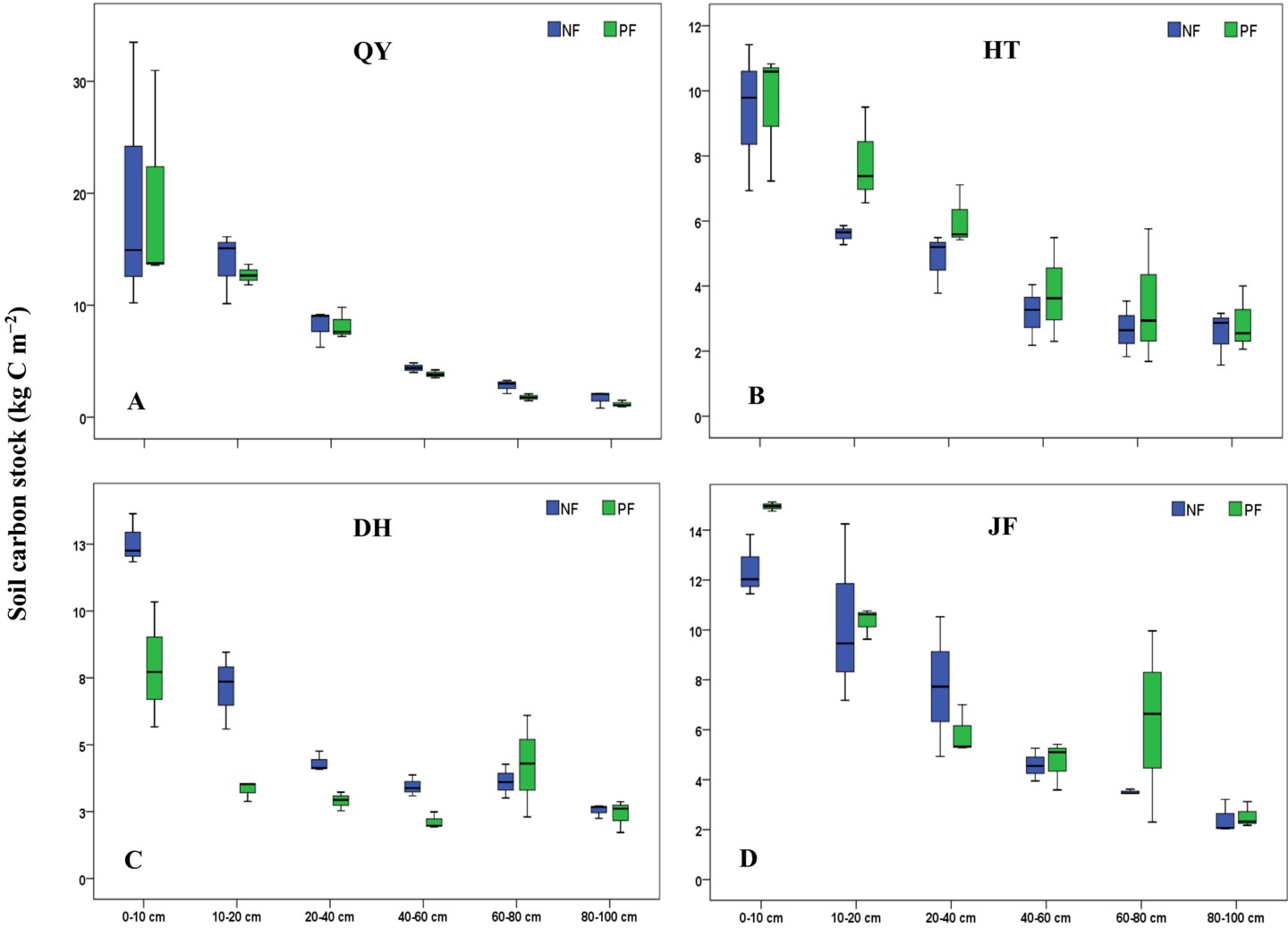

**Figure 2** **Soil C stocks under different forests in the 0–100 cm soil layers along the study sites and land use.** NF, Natural forest; PF, Plantation forest; (A) QY, Qingyuan; (B) HT, Huitong; (C) DH, Dinghushan; (D) JF, Jianfengling. Error bars are the standard errors of the means ($n$ = 3).

## RESULTS

### Soil C and N stocks

For all the sampling sites evaluated, soil C stock ranged from 2.6 ± 1.1 Mg ha$^{-1}$ to 38.6 ± 23.1 Mg ha$^{-1}$ and soil nitrogen stock ranged from 0.2 ± 0.1 Mg ha$^{-1}$ to 3.3 ± 1.5 Mg ha$^{-1}$. Following the land use change (LUC) in the two depth layers, a gain of soil C was observed at HT (+16%) while a loss of C was observed at QY (−7%), DH (−26%) and JF (−13%) (Fig. 2), similar result was found for N stock. The current study reported a loss in depth layers at QY (−8%), DH (−19%) and JF (−12%) (Fig. 3). The pattern distribution of N stock at HT site changed following the soil depth. A loss of −5% was observed in the topsoil while a gain of +24% was reported in the subsoil. Moreover, the Northeast region (QY) had the highest C and N stock followed by the Southeast (JF) (Figs. 2 and 3). Generally, the spatial distribution of C and N stock increased from South to North with the

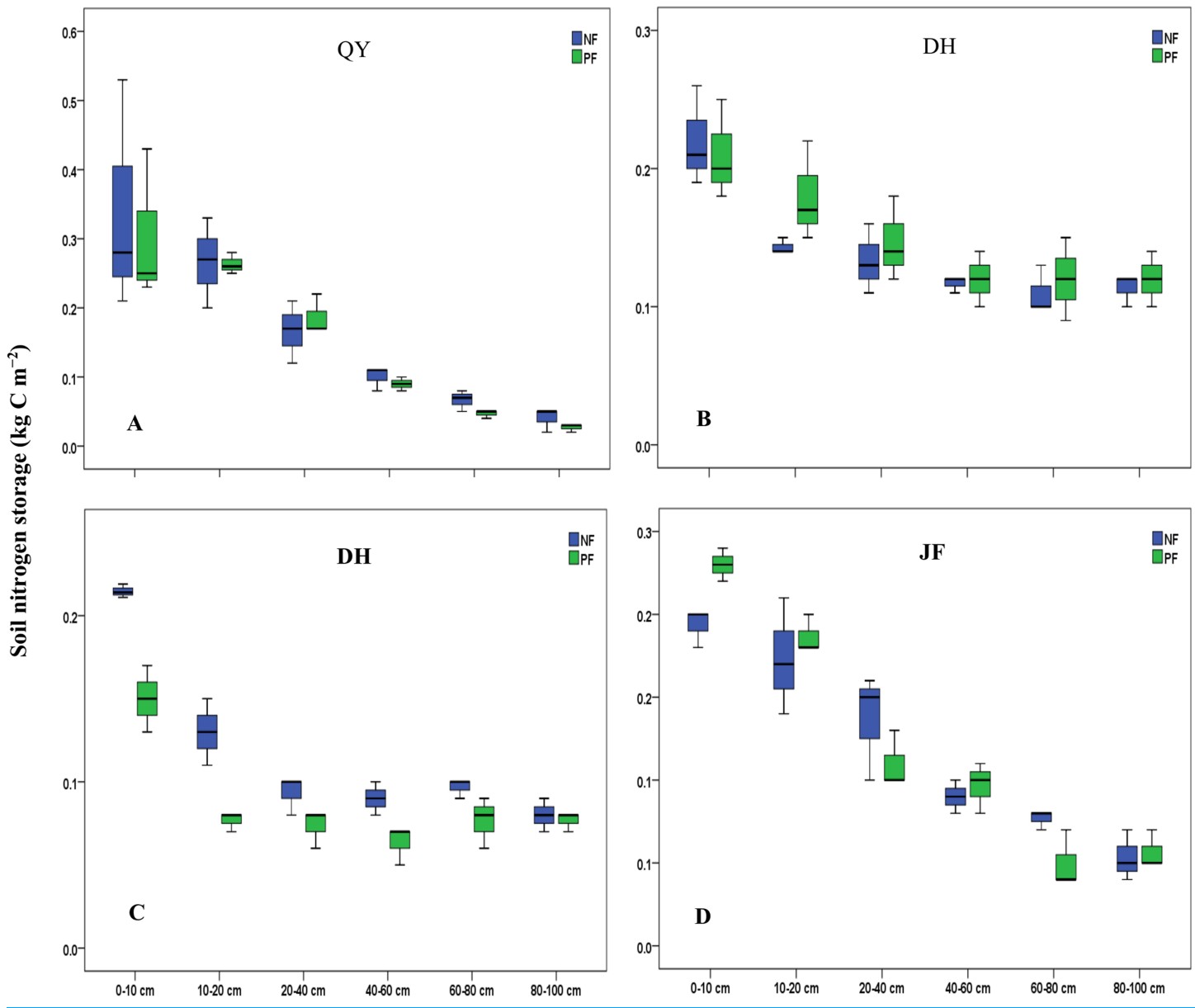

**Figure 3  Soil N stocks under different forests in the 0–100 cm soil layers along the study sites and land use.** NF, Natural forest; PF, Plantation forest; (A) QY, Qingyuan; (B) HT, Huitong; (C) DH, Dinghushan; (D) JF, Jianfengling. Error bars are the standard errors of the means ($n$ = 3).

decreasing climate factors. Soil C and N stock was significantly ($P < 0.001$) higher in the topsoil compared to subsoil in both forest types. The concentration of C in soil among 0–20 cm layers varied in the order of QY > JF > DH > HT in NF and in the order of QY > JF > HT > DH in PF. Similar result was observed for N stock. The highest and lowest concentrations of N stock were found at QY, respectively in both forest types.

## Relationship between soil C, N stocks, and underlying factors

Soil C and N stock varied significantly ($P < 0.05$, for all) with sites, soil depth under NF and PF but not their interactions (Table 3). The statistical analysis showed a strong correlation

**Table 4 Pearson's coefficients correlation between soil C, N stocks and affecting factors.**

| Depth (cm) | Forest type | Variable | pH | BD (g cm$^{-3}$) | Age (years) | Elevation (m) | MAT (°C) | MAP (mm) |
|---|---|---|---|---|---|---|---|---|
| 0–10 | NF | C stock | 0.367 n.s | −0.628* | / | 0.098 n.s | −0.625* | −0.462 n.s |
|  |  | N stock | 0.349 n.s | −0.584* | / | 0.227 n.s | −0.623* | −0.371 n.s |
|  | PF | C stock | −0.065 n.s | 0.016 n.s | 0.588* | −0.591* | 0.433 n.s | 0.082 n.s |
|  |  | N stock | 0.205 n.s | −0.270 n.s | 0.154 n.s | −0.537* | −0.478 n.s | −0.689** |
| 10–20 | NF | C stock | 0.468 n.s | 0.766** | / | 0.944** | −0.893** | −0.803** |
|  |  | N stock | 0.424 n.s | 0.849** | / | 0.973** | −0.836** | −0.722** |
|  | PF | C stock | −0.707** | −0.612* | 0.068 n.s | 0.461 n.s | −0.255 n.s | 0.198 n.s |
|  |  | N stock | −0.733** | −0.607* | 0.518* | 0.251 n.s | −0.616* | −0.252 n.s |

Notes:
** Correlation is significant at the 0.01 level.
* Correlation is significant at the 0.05 level.
n.s, not significant; $n = 15$; BD, Bulk density; NF, Natural forest; PF, Plantation forest; MAT, Mean annual temperature; MAP, Mean annual precipitation.

between soil C and N stock ($R^2 = 0.93$, $P < 0.001$ and $R^2 = 0.71$, $P < 0.001$ for NF and PF, respectively). However, soil pH under the NF did not significantly change in all depths compared to that under PF which was negatively correlated with soil C and N stock in the subsoil of PF. In general, the BD increased progressively with increasing depth in both depth layers and did not have a strong effect on soil C and N stock in the topsoil of PF (Table 2). Plantation age and elevation were positively corelated with soil C and N except at 0–10 cm of PF where elevation was negatively correlated with their values (Table 4).

### Relationship between soil C, N stocks and climate factors

Soil C stock increased with increasing N stock in NF and PF LU types (Fig. 4). Moreover, we observed a significant relationship between these variables in NF ($R^2 = 0.93$, $R^2 = 0.71$, $P < 0.001$ for all in PF and NF, respectively). In general, soil C was negatively correlated to environmental factors following NF in both depth layer (Table 4), whereas soil N stock was significantly correlated with MAT in NF of both depth layers ($P < 0.05$, for all).

## DISCUSSION

Specifically, we focused our attention on soil C and N at the topsoil (0–20 cm) depth layer in different LU type because topsoils accumulate more C and N than deeper soil layers (*Deng et al., 2016*; *Angst et al., 2018*; *Zhang et al., 2019*) and the subsoil generally contains less soil C (*Rumpel & Kögel-Knabner, 2011*; *Kunlanit, Butnan & Vityakon, 2019*).

### Effects of land-use change on soil C and N stock

Our results indicated that LUC from NF to PF is one of the key variables explaining the variation in soil C and N stock. The statistical analysis showed that soil C and N stock was significantly different across forest types ($P < 0.01$). Although a gain of C and N was observed at HT, the current study showed a clear decrease by −7%, −13% and −26% at QY, JF and DH, respectively, following the conversion from NF to PF. These findings suggest that LUC from NF to PF influences C and N inputs into the soil followed by a decrease in soil C and N stocks, which is in line with previous studies. For example, *Guo & Gifford (2002)* reported a decrease in soil C stocks after conversion forest to plantation

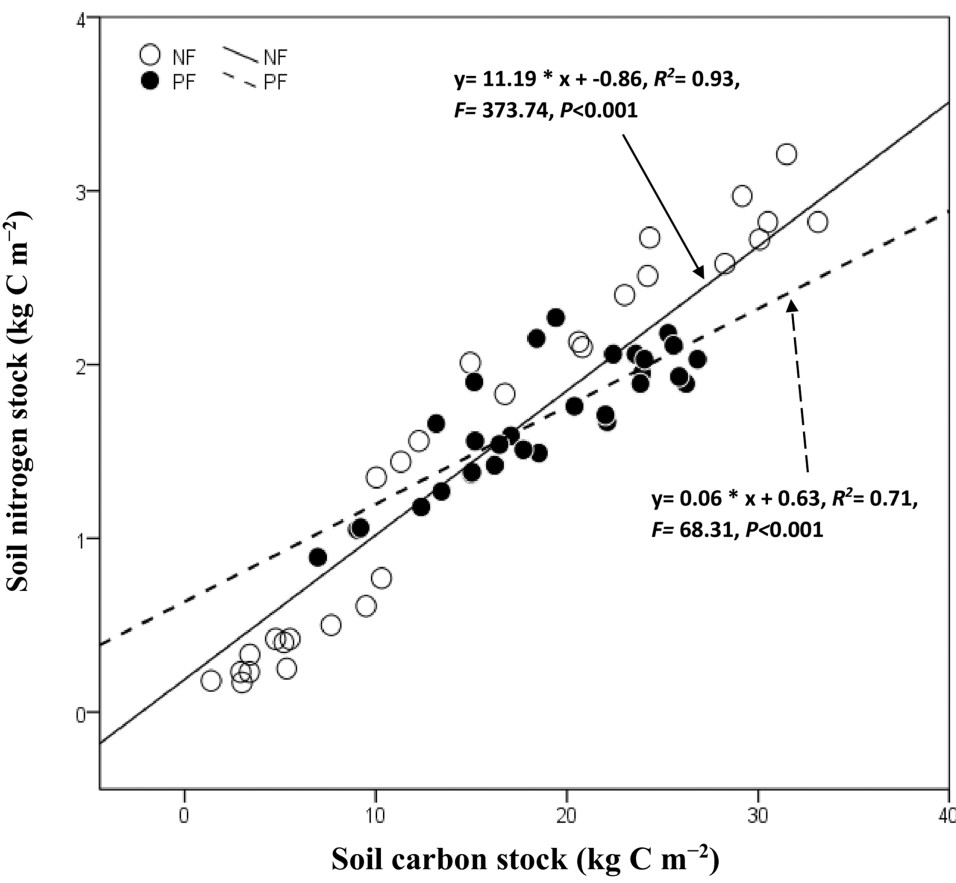

**Figure 4  Correlation between soil carbon stock and nitrogen following land use.** NF, Natural forest; PF, Plantation forest.        

by −13% whereas *Girmay et al. (2008)* observed a decrease by −83% in topsoil (0–10 cm). The patterns of gain and loss in soil C and N can vary according to a broad variety of soil types, shifts in abundance of woody and herbaceous vegetation, microbial activities, altered soil water, and temperature regimes which accelerate decomposition (*Covington, 1981*; *Johnson et al., 1995*; *Jackson et al., 2000*; *Turner, Lambert & Johnson, 2005*). The change of litter fall/input, the amount and type of plant residues associated with microbial activities produced by plants can explain, in part, the change of C and N stock. According to *Arevalo et al. (2009)*, the lack of mixing mineral soil with the surface litter material for example, resulted in low C stocks and the accumulation of litter in the humus layer. On the other hand, soil N stock is attributed with the change of N input such as atmospheric N deposition, biological N fixation and output such as N uptake by plant, N emission to groundwater or the atmosphere (*Li, Niu & Luo, 2012*).

Changes in soil C and N are associated with changes of trees species and their diversity through variation in litter quantity and quality, exudates and turnover rate of roots (*Wang et al., 2013*; *Hoogmoed et al., 2014*; *Deng & Shangguan, 2017*). Our findings largely confirm this trend as it was observed that *Pinus koraiensis* stock five times more soil C than *Metasequoia glyptostroboides* and two times more than *Cunninghamia lanceolata* in

PF. Similar results were observed for N stock. In addition, *C. lanceolata* stocks four times more N than *M. glyptostroboides*. Variation in soil C and N values with tree species as demonstrated in the present study has been also reported previously. For example, *Demessie, Singh & Lal (2011)* argued that coniferous species (*Pinus patula* and *Juniperous procera*) accrue more stock of C and N compared to the Eucalyptus species. Other studies suggested that soil C stock was generally larger under coniferous species than broadleaf species (*Augusto et al., 2002*; *Kasel & Bennett, 2007*; *Schulp et al., 2008*) or increases linearly with tree species diversity (*Montagnini & Porras, 1998*). According to *Wang et al. (2013)*, SOC stock in 0–20 cm layer was significantly higher in the mixed plantation than in the monoculture plantations. Thus, the potential of C and N sequestration in soil varies from one species to another within an area.

## Effects of soil depth on soil C and N stock

In general, soil C and N stocks decrease with increase of soil depth. For example, the topsoil of C accounts for 59% (QY) and 57% (JF) in NF, and similar results were observed in PF. In addition, soil N stock accounted for 68% in NF and 53% in PF of HT which is consistent with previous studies. Similar to our results, *Li et al. (2019)* and *Wang et al. (2016)* found a decrease of soil C and N stocks with depth. Therefore, we proposed the hypothesis that topsoil is most biologically active than subsoil and thus, has higher soil C and N accumulation. In addition, our results also provide some indication that soil C and N stock do not only reduce in the topsoil (0–10 cm), but also in the subsoil (10–20 cm). The statistical analysis showed a significant variation and negative correlation of soil C and N following soil depth indicating that soil C and N sequestration mechanisms varied with the soil profile. This result could be due to destruction of the existing litter layer and an increasing soil organic C mineralization rate through to the low rate of canopy covers which exposed the soil in PF to the solar radiation (*Chen & Wang, 2007*). On the other hand, soil disturbance on vegetation can also, in part, explain this result by way of differences in root distribution and its management. Obviously, different vegetation types will have different litter decomposition processes consequently leading to differences in release of C and N in the soil (*Zhang et al., 2013*). *Jobbágy & Jackson (2000)* for example, argued that root distributions affect the vertical placement of C in the soil, and above- and below-ground allocation affects the relative amount of C that eventually falls to the soil surface from shoots. Moreover, the action of deep roots which create pores and facilitate the movement of this nutrient could explain the larger soil N stock observed in NF (*Poirier, Roumet & Munson, 2018*). In addition, vegetation management influences the balance of forest C by desired crop tree species and by increasing the rate of C storage and biomass accumulation (*Colombo et al., 2005*). Consequently, in the current study soil disturbance affected the balance of C entering via plant exudates and residues and C output through mineralization in the soil and on N stock.

The variation in soil C and N with soil depth likely results from type of soil as postulated in other studies. *Da Silva Santana et al. (2019)* proposed that C and N stocks was lower in Planosols than Acrisol and Ferralsol in Brazil partly due to their shallowness and their lower capacity to stabilize N in the organic form. *West & Six (2007)* further suggested

that soil type influenced the equilibrium between the C inputs and outputs. The current study showed that the level of C and N stock in lateritic yellow soil (Ultisols in the US Soil Classification) of JF was significantly higher with those found in mountain reddish yellow earth of HT site. However, no significant difference has been observed between the soil type at DH and JF. This finding indicates that the residence time and capacity of sequestration varies with soil types.

## Effects of BD and soil pH on soil C and N stock

The statistical analyses showed that soil BD and pH vary significantly with soil depth. Soil pH tends to decrease with an increasing soil depth under different forest types while the opposite has been observed for BD. Our study showed that BD was strongly correlated with soil C and N stock except in the topsoil of PF, but only significantly correlated with pH in the subsoil of PF . This result suggests that the pattern distribution of soil BD and pH affects soil function such as soil microbial community and microbial activity (*Thomas, 1996*) which are closely related to soil C and N stock. It has been suggested that low soil pH can decrease microbial biomass and activity (*Blagodatskaya & Anderson, 1998*) and might also lead to the accumulation of soil C (*Beets, Oliver & Clinton, 2002*, *Chen, Xu & Mathers, 2004*).
In addition, soil pH is influenced by trees species that can directly affect the pattern distribution of soil C and N stock since tree species have different capacity of sequestration in the soil. Generally, a low BD at total depth of 0–20 cm in both forest types was observed and it was significantly influenced by type of LU and not by soil depth. The lowest rate of BD was observed in DH where curiously the lowest values of soil C and N stock has been observed. This finding prompts us to hypothesize that the rate of BD may influence C and N input in the soil. Consistent with our findings, *Demessie, Singh & Lal (2011)* suggested that the lowest C and N stocks compared to the reference under *Cupressus lusitanica* compared to the other plantation sites may be partly ascribed to the lower BD along the profile.

## Effects of elevation and stand age on soil C and N stock

The current study showed that elevation is one of the main factors controlling soil C and N stock variation. We observed that elevation favors C and N accumulation in soils particularly in both depth layers of NF while it had a negative correlation in the topsoil of PF. *Saby et al. (2008)* found that elevation was a control factor on SOC in a French region. Previous studies reported a lower carbon and nitrogen stock at lower elevation (*Tesfaye et al., 2016*). This trend was confirmed in the current study as the lowest values in soil C and N stock were found at DH which is located at the lowest elevation (275 m). Besides, the pattern of distribution of soil C and N stock following the elevation showed an increase of those variables with an increase in elevation. This finding is in line which the results obtained by *Jones et al. (2005)* who reported that the largest SOC stocks occur in high elevation areas of Europe.

The mechanism of soil carbon sequestration with vegetation restoration is more complex (*Wang et al., 2019*). Although it has been reported that there is a high rate of growth and carbon uptake in young trees, our finding reported a higher soil C and N stock in NF compared to PF yet it has younger vegetation. This result could be explained by the

stand age. In the current study, NF has much longer stand age than PF, hence soil C and N inputs have probably more time to accumulate in the soil of NF through litter and roots in the long term compared to PF. Our findings are therefore consistent with the results found by *Arevalo et al. (2009)* who reported an increase of SOC stock with plantation age. In addition, our results showed that plantation age positively affected C and N stock with time. Interestingly, in PF, the highest rate of C and N stock was found at the QY which is the oldest PF site with 33 years old and the lowest values were observed at the youngest site of DH. Consequently, older stand general has a higher soil C stock because of the long-term soil C accumulation (*Sariyildiz, Savaci & Kravkaz, 2015*). *Deng et al. (2017)* reported that soil C stocks of *Caragana korshinskii* plantations increased remarkably with stand age from young to mature plantations in the Loess Plateau, whereas *Laganière, Angers & Paré (2010)* reported a gain of 6.1% and 18.6% in mature stage (10–30 years) and older stage (>30 years) plantations, respectively, with increase of time. Therefore, PF might need the same time or longer time period to accumulate the same level of NF in soil C and N stock in the soil. However, the verification of this hypothesis could be difficult because C stored in NF has a longer residence time and has a greater susceptibility to loss in PF (*Mackey et al., 2008*). Moreover, the difference observed could be also due to site management such as site preparation, the effects of harvesting, the type and level of silviculture activities and the antecedent soil fertility.

## Effects of forest management on soil C and N stock

Forest management can also affect the change of soil C and N stock through soil erosion or deposition. Certain management practices such as the site preparation by breaking down the physical protection of soil C can significantly increase the decomposition of soil organic C (*Guo & Gifford, 2002*). Thus, increasing the frequency of erosion after rainfall consequently influences soil structural stability and porosity. Our finding indicates that although it has been demonstrated that SOC does not accumulate indefinitely (*Johnston, Poulton & Coleman, 2009*), this alternative of replacement of NF with PF can be effective if the right measures are implemented avoiding as site preparation with burnt treatment, soil erosion, vegetation burning, wood products were harvested and (/or) an increased output as plantations (*Harmon, Ferrell & Franklin, 1990*; *Berthrong, Jobbagy & Jackson, 2009*; *Liao et al., 2010*) and maximizing litter inputs to soils. Otherwise, it will be not a sustainable measure because its performance can decrease over time partially if proper management actions are not taken. *Luo & Zhang (2006)* for example, reported that soil organic C stock decreased by 10% from the first to the second rotation for *C. lanceolata* plantations, and by 15% from second to the third rotation. The application of good management practices in PF could run up soil C and N sequestration after several years through maximizing litter inputs and might be valued to gain of C and N storage in the soil. The low values in soil C and N stock observed at DH could be as a result of the high level of site disturbance in this site which considerably decreases the rate of fine root biomass and affects the balance between biomass production and decomposition, root distribution or vegetation communities. According to *Jobbágy & Jackson (2000)* root distributions affect the vertical placement of C in the soil and *Yimer, Ledin & Abdelkadir (2006)* reported

in the Bale Mountains that the mean SOC stocks were lower among the vegetation communities in the western and northern aspects than in the southern and the eastern aspects both in the upper 0.3 m soil layer.

## Effects of climate factors on soil C and N stock

Soil C and N stocks are influenced by the complex interactions of climate (*Lal, 2005*) due to its impact on microbial activity and forest growth, consequently on the quantity and quality of organic residue soil inputs and on the rates of soil organic matter mineralization and litter decomposition (*Quideau et al., 2001*; *Heviaa, Buschiazzoa & Heppera, 2003*). Higher soil temperatures for example, increase microbial decomposition of organic matter (*Conant et al., 2011*) and high precipitation can also lead to C transport down the soil profile as dissolved and/or particulate organic matter (*Borken & Matzner, 2009*). According to *Post et al. (1982)*, the influence of climate factors results from their influence on the balance of carbon inputs from plant production and outputs through decomposition in soil. The current study reported a significant impact of climate factors on the soil C and N stock in both depth layers. Soil C and N was negatively linked with climate factors; we observed a decrease in soil C and N stocks with MAT and MAP following soil depth layer. This finding is consistent with previous studies. *Vieira et al. (2011)* for example, reported a significant inverse correlation between carbon and nitrogen stocks and soil temperatures whereas *Quideau et al. (2001)* and *Heviaa, Buschiazzoa & Heppera (2003)* argued that this is probably due to their effects on the rates of soil organic matter mineralization and litter decomposition and on the quantity and quality of organic residue soil inputs.

Previous studies reported a decrease in SOC levels followed by an increase of MAT (*Wang et al., 2004*), a trend that was partially confirmed in our study. The highest values of soil C and N stock were found at the coldest area, particularly at QY. However, we also observed an increase in soil C and N stock from Southeast to Northeast, particularly from JF to QY. The highest values were found at QY probably due to lower temperature conditions favoring accumulation of organic matter in these vegetation communities (*Post et al., 1982*).

Interestingly, we observed a similar pattern of distribution between soil C and N stock and climate factors. Soil C and N stock decreased with decreasing precipitation and temperature from the Southeast to the Northeast. This trend is in line with previous studies (*Jobbágy & Jackson, 2000*; *Amundson, 2001*; *Homann, Kapchinske & Boyce, 2007*), *Townsend, Vitousek & Trumbore (1995)* and *Trumbore, Chadwick & Amundson (1996)* who indicated that lower temperatures could result in reduced SOC breakdown, thereby increasing SOC accumulation. By contrast, we observed that the highest soil C and N were found at QY under the lowest precipitations followed by JF which was under the highest precipitations. This differential response could be attributed to low decomposition of litter under those precipitation regimes (*Seneviratne, Van Holm & Kulasooriya, 1997*) and the complex interactions which exist between plant species, soil conditions, microorganisms and climatic conditions. According to *Leifeld, Bassin & Fuhrer (2005)*, it could be due to the couple effect of lower temperatures and higher altitudes which

probably limit C turnover, which results in increased C accumulation even under conditions of smaller productivity and C inputs. Although, it is generally accepted that MAT and MAP temperature are the main factors influencing the potential of C and N sequestration, we could not find consensus on the specific effects of climate factors in soil C and N.

### Correlation between C and N stock

Similar to previous studies (*Li, Niu & Luo, 2012*; *Deng, Shangguan & Sweeney, 2013*; *Deng et al., 2016*), we observed a significant positive correlation between soil C and N stock in both forest types. The spatial distribution of soil C and N stock was similar with the largest values observed in the Northeast (QY) followed by Southeast (JF). We also observed an increase of N stock following an increase of C stock, indicating that the C budget is limited by the availability of soil N due to the coupling effect between C and N cycles in forest ecosystem (*Reich & Oleksyn, 2004*; *Melillo et al., 2011*). Besides, the relationship of soil C and N stock in NF was significantly stronger in the topsoil than in the subsoil probably due to the effect of rainfall on N vertical distribution. According to *Deng et al. (2016)*, rainfall can facilitate the migration of N into deeper soils, increase N accumulation in the subsoil during revegetation process and vegetation restoration, and thereby decreased soil C–N relations. This finding provides support to the trend observed below, N dynamics regulate terrestrial carbon sequestration for example, increasing N inputs lead to sustainable C sequestration (*Deng et al., 2016*). Thereby, high soil N concentration stimulates tree growth, which potentially increases carbon inputs into soils through litterfall and rhizo deposition, and promotes SOC sequestration by decreasing decomposition rates of old litter and recalcitrant soil organic matter by suppression of soil microbes and by chemical stabilization (*Jandl et al., 2007*; *Mo et al., 2008*).

## CONCLUSIONS

Land-use changes from NF to PF significantly affected soil C and N input although controlling factors induced differences in C and N stocks. The destruction of NF contributes to the loss of C and N from soil and a net increase of carbon in the atmospheric $CO_2$, which aggravates climate change. Our study demonstrated that although PFs can lead to higher C and N sequestration and have been promoted as a measure to mitigate future climate change, NFs have below-ground C and N processes which promote better accumulation in the soil. Although PF is one obvious approach of maintaining or increasing future wood supply and mitigating the impacts related to their destruction, it cannot replace the ecological roles played by NF especially if we consider animal biodiversity and the loss of their habitat. The best way to protect and preserve NFs for a healthy environment is to encourage the reforestation of degraded environments. The results of the currents study showed also the complex interaction existing between abiotic and biotic factors and soil C and N input.

## ACKNOWLEDGEMENTS

Authors thank Karanja Joe, Ntambo Mbuya and Mensah Raphael for their constructive suggestions on this article.

### Funding

This study was sponsored by the National Natural Science Foundation of China (No. U1805243), State Key Laboratory of Soil and Sustainable Agriculture and China Scholarship Council (CSC) University Scholarship Program. The funders had no role in study design, data collection and analysis, decision to publish, or preparation of the manuscript.

### Grant Disclosures

The following grant information was disclosed by the authors:
National Natural Science Foundation of China: U1805243.
State Key Laboratory of Soil and Sustainable Agriculture and China Scholarship Council (CSC) University Scholarship Program.

### Competing Interests

The authors declare that they have no competing interests.

### Author Contributions

- Mbezele Junior Yannick Ngaba conceived and designed the experiments, performed the experiments, analyzed the data, prepared figures and/or tables, and approved the final draft.
- Xiang-Qing Ma conceived and designed the experiments, performed the experiments, authored or reviewed drafts of the paper, and approved the final draft.
- Ya-Lin Hu conceived and designed the experiments, authored or reviewed drafts of the paper, and approved the final draft.

### Data Availability

Raw data is available as a Supplemental File.

### Supplemental Information

Supplemental information for this article can be found online at http://dx.doi.org/10.7717/peerj.8377#supplemental-information.

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
