# Peer review of "Variability of soil carbon and nitrogen stocks after conversion of natural forest to plantations in Eastern China"

_PeerJ, doi:10.7717/peerj.8377_

## Round 0.1 · original submission · Major Revisions

Please revise the manuscript carefully and address all comments of the reviewers.

Reviewer 1 ·

Basic reporting

The authors addressed an interesting topic that soil carbon and nitrogen stock responded to forest conversion in different altitudes. Generally, this is a traditional issue how plantations change soil carbon and nutrient pool. They found natural forests have a better capacity to accumulate soil carbon and nitrogen. Although this conclusion is not quit novelty, the paper provided rich data in understanding the relationships between afforestation or reforestation and soil carbon pool dynamics.

Experimental design

The description of methods is extremely poor. Line 99-100, the authors declared “six depth layers 0–10, 10–20, 20–40, 40–60, 60–80 and 80–100 cm along soil profiles”, but I can only see the results from two layers. Moreover, no detail information was given in lab-determining elements and other parameters. No sample site descriptions and any other detail information to tell reader about reasonable or statistical sampling. I can not judge the scientific property of sampling.

Validity of the findings

The results did not follow the methods. The interesting results is carbon stoke distribution along soil profile, especially the deep soil. However, I can only see the top soil. Line 140 “Plantation age and elevation were positively correlated with….”, if this is the case, how many plantations with different age or different elevations in a sample region? I don’t think “land use change” is a good phrase to express forest conversion.

Additional comments

Many abbreviations have no primary words, which limited the readable of whole paper. Scientific hypothesis should be listed in the section of Introduction. Most part of the paper paid more attention to carbon, less attention to nitrogen. Line 91-93, the three questions are too normal. Every papers about this direction can give these questions. Some deep objectives should be listed.
I think if the authors can add the detail information about materials and methods, and carbon and nitrogen stock along whole soil profile 0-100. If do so, the paper could be acceptable.

Reviewer 2 ·

Basic reporting

This paper investigated differences between the natural forests and plantation forests at four sites and tried to explore the factor affecting these differences. The experimental design and field measurements seem no problem. But I have some concerns with the current version of the manuscript. Particularly, the discussion section requires a thorough revision. Some conclusions and explanations are also very far-fetched. The authors should revise it and make it much clearer.

Experimental design

L27 five sites, L97 six forest sites. So how many forest sites did you have?
L97: you only use the 0-10 cm and 10-20 cm soil data in this paper. So even though you collected six layers soil samples, I think these could not be included in the total samples 252.
Does all of the samples is taken at the same time?
You should point out the layers of topsoil and subsoil.
L106-107: it’s better to use the formula editor to generate it.
Table 2: As far as my knowledge goes, the age of Dinghushan plantation forest is at least 60 years, you should check it again.

Validity of the findings

It seems no problems.

Additional comments

L30: which one was decreased?
L36-39: This sentence is too long.
L39: Please add the abbreviation of MAT and MAP.
L40: the first letter of the word should use upper case or lower case consistently.
L44: should be nitrogen
L46-48: I think you should state the reason why the soil N accessibility is limited. The logic is not right.
L69-71: Land use change, reforestation, afforestation, land conversation and forest management, they are different concept. You should make sure which kind of it in your paper and don’t mix it up.
L51-52, L63-67, L84-86: These sentences represent the similar meaning. Please revised it carefully.
L88: it should be “it’s”. it should be “plantation forests”.
L93: replace by How
L97: Although we know the meaning of PF. You should explain the abbreviation because it is the first appear.
L115: check citation for SPSS version 20.0. Systat Statistical Software Package for Windows?
L124: Remove “interestingly”
L121-124: which figure or table showed this result?
L129-130: this sentence belongs to the next section.
L135: I think you can separate this section into two parts or you can put the results of changes in affecting factor into the l section, this may be better. This results portion is some kind of simple.
L136: This result “Soil C and N stock varied significantly (P<0.05, for all) with site” belongs to the last section.
L156-162, L167-169: these sentences are a little repetitive.
L170: change the word of “prediction”
L180: It should be “Wang”
L170-172: the ages of this plantation forests are different, so it meaningful to compare these together?
I recommend to use the annual accumulate rate if you want to compare different species function.
L184-186: what does these percentage mean?
L188-189: the hypothesis should be proposed in the introduction section.
L193-195: This explanation is very far-fetched.
L218-219: How can you get this conclusion?
L246: this data (189 m) is not matched with the Table 2 (275 m).
L359-360: “forests have below-ground C and N processes which promote better accumulation in the soil”. How about the plantation forests?
The discussion part should also be revised with more explicit explanations of your findings.

Reviewer 3 ·

Basic reporting

The language needs to be improved. Generally speaking, academic papers use the present or past tense.

Experimental design

no comment

Validity of the findings

no comment

Additional comments

Line 96: The authors collect a total of 252 samples from NF and PF stand of each site in August 2017. Six forest sites were selected as experimental plots including Jianfengling (JF), Dinghushan (DH), Huitong (HT), and Qingyuan (QY), across eastern China forest (Fig. 1, Table1).
(1) Please give more details about collection area.
(2) How many samples chose from each forest site, and give the approximate location?
(3) 252 samples are sufficient for this article to get the conclusions. Please explain their rationality.

Line 145 In the discussion, the authors investigate the effects of different factors on soil C, N stock, respectively. The study provides an overview of the current spatial distribution and soil stocks of C and N, as well as the effects of environmental factors on soil C and N stocks and indicated that although MAT and MAP are the most important factors affecting the variations in soil C and N, their vertical and horizontal distribution differed in various aspects.
(1) However, in the natural environment, the soil C, N stock is affected by the multi-factors. If possible, the authors can discuss more factors.
(2) “ MAT and MAP are the most important factors affecting the variations in soil C and N”
How do they affect the soil C, N stock? Please try to explore the mechanism of synergism.

---

## Round 0.2 · accepted · Accept

The comments of the reviewers were addressed in a good way.